# Cholecystocutaneous Fistula

**DOI:** 10.3390/diagnostics14131386

**Published:** 2024-06-29

**Authors:** Francisco Tustumi, Giane Prata da Costa Filha, Guilherme Carvalhal Gnipper Cirillo

**Affiliations:** 1Department of Health Sciences, Hospital Israelita Albert Einstein, Sao Paulo 05652-900, Brazil; 2Department of Gastroenterology, Universidade de São Paulo, Sao Paulo 05508-220, Brazil

**Keywords:** healthcare disparities, health inequities, gallbladder, fistula

## Abstract

This image article presents an 88-year-old indigenous woman with a history of several episodes of abdominal pain, for which she went to numerous different hospitals in the countryside, was always treated with analgesics, and then discharged. After a long time, the patient eventually was evaluated with magnetic resonance imaging. The test revealed a displaced gallbladder with thickened walls, multiple stones, and a fistulous tract extending to the skin. This case underscores the significant challenges faced by patients in regions with limited healthcare access, highlighting the impact of delayed diagnosis and inadequate management on patient outcomes.

Appendix A. Magnetic resonance imaging showing a cholecystocutaneous fistula. An 88-year-old indigenous woman from the Brazilian countryside was referred for surgical evaluation due to a small lump in the anterior abdominal wall, with the main diagnostic hypothesis being a sebaceous cyst. This lump emerged years ago, and it used to become painful and inflamed on some days. Additionally, the patient reported that, occasionally, some stones emerged from this lump. The patient had no comorbidities. However, the patient had a history of several episodes of abdominal pain, for which she went to numerous different hospitals in the countryside, was always treated with analgesics, and then discharged. She also reported occasional fever. The intensity of symptoms had decreased in the last few years. She had never undergone any imaging test. The results of serum tests were unremarkable. After surgical team evaluation, the patient underwent magnetic resonance, which showed a displaced gallbladder with diffusely thickened walls, multiple stones, and a fistulous tract extending from the gallbladder to the skin of the anterior abdominal wall. In addition, multiple confluent retroperitoneal enlarged lymph nodes were found in this exam. The patient was referred for hematological consultation to investigate retroperitoneal lymph node enlargement. The patient died of pneumonia some months later without treating the cholecystocutaneous fistula. The cholecystocutaneous fistula usually affects elderly women, with an average age of 72 years, with poor socioeconomic conditions and poor access to the health system [1,2]. Although the definitive treatment for this condition is a cholecystectomy with fistula excision, high-risk patients may experience treatment delays [2,3]. Vulnerable patients face inefficient and slow healthcare, and the time required for diagnosis and treatment planning can extend over years. Consequently, the patient remained untreated, enduring symptoms and an impaired quality of life. This case underscores the significant challenges faced by patients in regions with limited healthcare access, highlighting the impact of delayed diagnosis and inadequate management on patient outcomes (Figure 1).

## Figures and Tables

**Figure 1 diagnostics-14-01386-f001:**
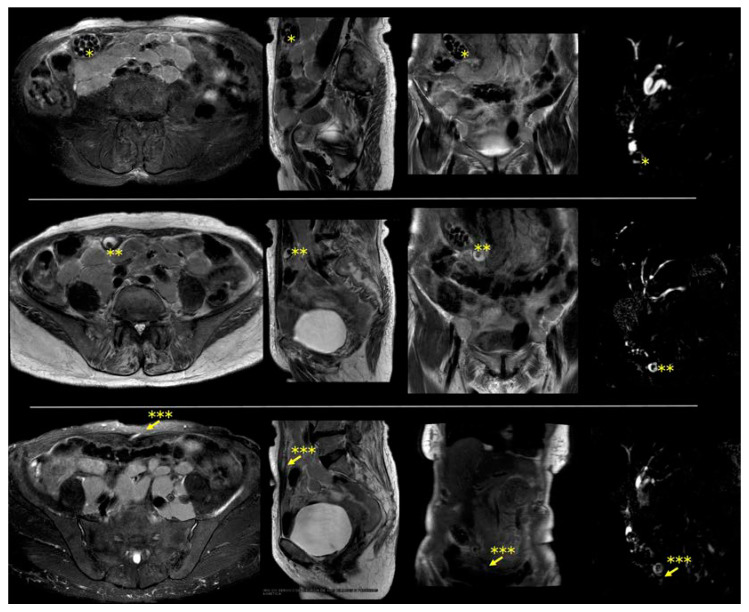
Magnetic resonance imaging shows a caudally displaced gallbladder with numerous stones and with cholecystocutaneous fistula. Left to right: axial, sagittal, coronal, and cholangiography. * Displaced gallbladder; ** stone inside the fistula; *** cholecystocutaneous fistula (indicated by arrow).

## Data Availability

No new data were created or analyzed in this study. Data sharing does not apply to this article.

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
