# Peer review of "Cholecystocutaneous Fistula"

_diagnostics, 2024, doi:10.3390/diagnostics14131386_

Round 1

Reviewer 1 Report

Comments and Suggestions for Authors

Cholecysto-cutaneous fistulas are rare but could be managed properly today. The author did point out the problem of delayed diagnosis due to the lack of healthcare resources. The supplementary MP4 is clear. However, the authors need to highlight the cholecysto-cutaneous tract in the images of the manuscript with an arrow. In addition, the authors better showed the fistula in the lateral sagittal view.

Author Response

Carol Zhang, Editor

Diagnostics

Dear Ms. Zhang,

I appreciate the opportunity to submit our manuscript titled "Cholecystocutaneous fistula" (Manuscript ID: diagnostics-3076239) to Diagnostics. We have carefully considered the feedback provided by the reviewers and have made the following revisions to improve our manuscript.

Reviewer 1:

Comment: The authors need to highlight the cholecysto-cutaneous tract in the images of the manuscript with an arrow.

Response: We have revised the images in the manuscript to include arrows that highlight the cholecysto-cutaneous tract as requested.

Comment: The authors better showed the fistula in the lateral sagittal view.

Response: We have included additional images showing the fistula in the lateral sagittal view to provide a clearer depiction of the condition.

We believe that these revisions address the concerns raised by the reviewers and improve the overall quality of our manuscript. We hope that the revised manuscript now meets the standards for publication in Diagnostics.

Thank you for considering our revised submission.

Sincerely,

Dr. Francisco Tustumi

Reviewer 2 Report

Comments and Suggestions for Authors

Dear authors, 

The Interesting Images is a very nice and interesting case. Images are of good quality and video are extremely useful. Video editing with label is perfect. I would suggest you to point out the findings also in the images and you should improve the figure legends adding sequences and explaining and pointing the findings.

Author Response

Carol Zhang, Editor

Diagnostics

Dear Ms. Zhang,

I appreciate the opportunity to submit our manuscript titled "Cholecystocutaneous fistula" (Manuscript ID: diagnostics-3076239) to Diagnostics. We have carefully considered the feedback provided by the reviewers and have made the following revisions to improve our manuscript.

Reviewer 2:

Comment: The authors should point out the findings in the images and improve the figure legends by adding sequences and explaining and pointing out the findings.

Response: We have revised Figure 1 and figure legend to include detailed sequences and explanations of the findings. We have also pointed out the relevant findings directly in the images to enhance clarity.

We believe that these revisions address the concerns raised by the reviewers and improve the overall quality of our manuscript. We hope that the revised manuscript now meets the standards for publication in Diagnostics.

Thank you for considering our revised submission.

Sincerely,

Dr. Francisco Tustumi